# Effect of Photofunctionalization with 6 W or 85 W UVC on the Degree of Wettability of RBM Titanium in Relation to the Irradiation Time

**Arturo Sanchez-Perez** [1,]*[iD], **Nuria Cano-Millá** [1], **María José Moya Villaescusa** [1], **José María Montoya Carralero** [1] **and Carlos Navarro Cuellar** [2]

[1] Department of Periodontology, Medicine and Dentistry Faculty, Murcia University, 30008 Murcia, Spain; nuria.canom@um.es (N.C.-M.); mjm.villaescusa@um.es (M.J.M.V.); jmmontoya@um.es (J.M.M.C.)
[2] Department of Surgery, Complutense University, 28040 Madrid, Spain; cnavarrocuellar@gmail.com
[*] Correspondence: arturosa@um.es; Tel.: +34-968-247-946





**Featured Application: Due to the inevitable ageing of titanium, the use of a reliable, fast, and inexpensive method to reverse the effects of time on implants is of clinical relevance.**

**Abstract:** Photoactivation with ultraviolet C light can reverse the effects derived from biological ageing by restoring a hydrophilic surface. Ten titanium discs were randomly divided into three groups: a control group, a 6 W group, and an 85 W group. A drop of double-distilled, deionized, and sterile 10 μL water was applied to each of the discs. Each disc was immediately photographed in a standardized and perpendicular manner. Measurements were taken based on the irradiation time (15, 30, 60, and 120 min). UVC irradiation improved the control values in both groups. There was no difference in its effect between the 6 W group and the other groups during the first 30 min. However, after 60 min and up to 120 min, 85 W had a significantly stronger effect. The contact angles with the 85 W ultraviolet light source at 60 and 120 min were 19.43° and 31.41°, respectively, whereas the contact angles for the 6 W UVC source were 73.8° and 61.45°. Power proved to be the most important factor, and the best hydrophilicity result was obtained with a power of 85 W for 60 min at a wavelength of 254 nm.

**Keywords:** titanium; dental implants; hydrophilic; hydrocarbon; biological ageing; UV photofunctionalization

## 1. Introduction

Titanium is a material devoid of toxicity and is stable and easy to obtain. Its good physical properties have led to its widespread use in oral implantology.

To improve its clinical characteristics, there have been attempts to increase the bioactivity of titanium implants [1–3]. The focus was on improving osseointegration, which is defined as the direct and functional connection between living, structured bone and the surface of an implant under load.

Among other factors, the physicochemical properties of the implant surface, its topography, and chemical composition have been the focus for improving biocompatibility [4]. Under optimal conditions, this improvement implies an increase in the absorption of ions and molecules, thus stimulating cellular attraction, proliferation, and expansion. The result is a higher degree of contact between the bone and the implant (BIC).

One method to establish surface improvement is through its wettability, which is based on the angle of contact between a droplet of water and the surface of the material being checked (Figure 1).

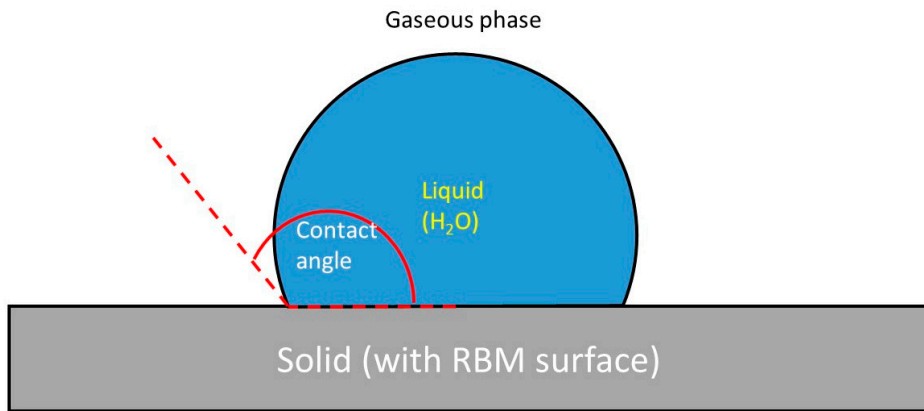

**Figure 1.** The wettability of a surface depends on the surface energy of the solid, the surface tension of the liquid, and the gaseous phase. Wettability can be established by measuring the contact angle between the liquid and the solid.

Arbitrary degrees of wettability have been established, depending on the angle of contact, and the parameter is classified into four categories: superhydrophobic, hydrophobic, hydrophilic, and superhydrophilic (Figure 2).

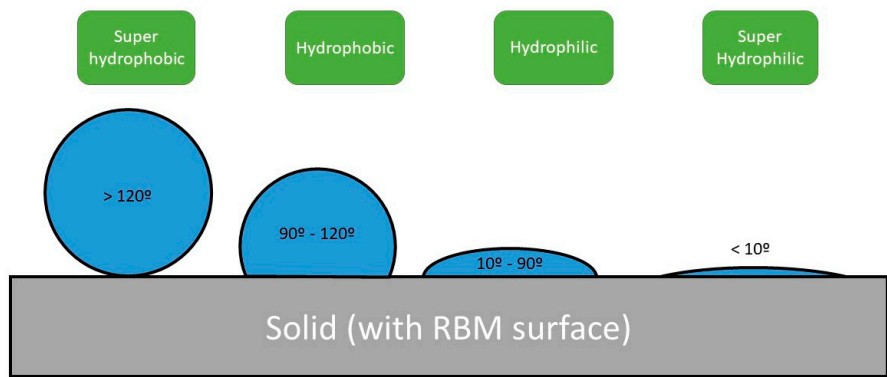

**Figure 2.** Classification of wettability as a function of contact angle. From superhydrophilic to superhydrophobic.

There are reports that titanium surfaces become contaminated by atmospheric carbon compounds being deposited on their surface, causing them to become progressively more hydrophobic. This process is called biological ageing and has been widely discussed in various works.

Over time, the percentage of carbon deposited on titanium increases from 14% to 63% with important clinical consequences.

Aged titanium surfaces exhibit 50% less absorption of amino acids, proteins, and osteoblast adhesion compared with newly manufactured titanium surfaces. Such alterations are considered unavoidable and depend on the storage time, having been detected as early as four weeks after manufacture.

The physicochemical effect produced by the application of ultraviolet light to the titanium surface is called photofunctionalization, with biological effects. Photofunctionalization is capable of creating a superhydrophilic surface without altering or modifying any other characteristic of the titanium surface. Therefore, it has been proposed as a simple and effective method to reverse the effects of titanium ageing, which can in turn lead to a faster and more complete establishment of bone-titanium integration. Some authors have called this phenomenon superosseointegration.

The purpose of our study was to determine the effect of two UV-C light sources with a wavelength of 254 nm and an energy of 1400 $\mu W/cm^3$ to obtain a hydrophilic surface, comparing two UV-C sources with different powers (6 and 85 W) and a control group.

## 2. Materials and Methods

### 2.1. Materials

The materials used in this study were as follows:

- Ten titanium discs 6 mm in diameter and 1 mm thick with biological ageing for 4 weeks;
- Ultraviolet light sources:
  - VL-6C (Analyzer, Murcia, Spain),
  - Ultraviolet germicidal sterilizer (Quirumed, Valencia, Spain);
- Ten titanium discs with biological ageing for 4 weeks;
- Nikon D 80 DSLR Camera (Nikon Corp., Tokyo, Japan);
- Sigma macro 110 mm lens (Kabushiki gaisha, Kanagawa, Japan);
- IMAGEJ image analysis program (National Institutes of Health, Bethesda, MD, USA);
- Micropipette Easy 10 KG 578545 (10 µL) (Labbox Labware, S.L., Barcelona, Spain);
- SPSS V23 statistical package IBM (SPSS Inc., Chicago, IL, USA).

### 2.2. Method

The 10 discs were stored for 4 weeks in darkness and in a temperature- and humidity-controlled environment. Each disc underwent the same process of biological ageing, after which they were randomly distributed into 3 groups using the Internet program random.org:

- Control group;
- 6 W group;
- 85 W group.

All the groups received one 10 µL drop of water from an Easy 10 KG 578545 pipette, which was photographed perpendicularly, and its angulation was measured using ImageJ. Measurements were recorded according to the irradiation time at 15, 30, 60, and 120 min.

In each group, one disc was used as a control, which received no irradiation and was processed according to the same methodology (Figure 3).

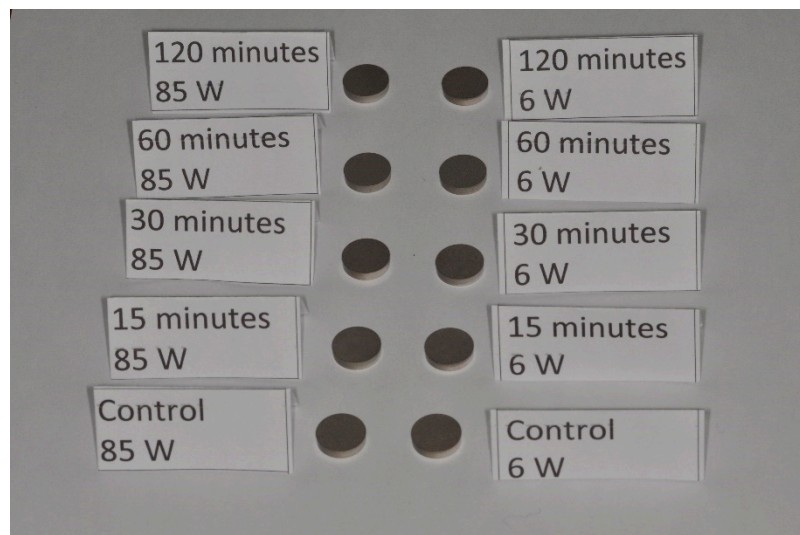

**Figure 3.** Distribution of discs into groups irradiated with 6 or 85 W according to irradiation time.

Statistical analysis was performed using IBM SPSS ver. 23.0 (IBM Co., Armonk, NY, USA). A normality assumption for the data was tested using the Shapiro–Wilk test.

Descriptive statistics were used to synthesize the collected data. Mean comparisons within the group were computed using analysis of variance, whereas median differences between the groups were analysed using the Mann–Whitney U-test. Statistical significance was considered when $p \leq 0.05$.

All analyses were conducted by an independent statistician who was blind to the procedure performed (http://estadisticamurcia.com/web/#2) (accessed on 7 June 2021).

## 3. Results

All variables followed a normal distribution except the variable of 6 W and 15 min. The average contact angles for the control discs were 116.19° for the 6 W control group and 117.6° for the 85 W control group. The mean angles obtained at 15, 30, 60, and 120 min with the 6 W ultraviolet (UVC) light source were 69.66°, 74.38°, 73.8°, and 61.45°, respectively. When an ultraviolet (UVC) light bulb of 85 W was used as the source, the average angle obtained at 15 min was 70.25°, 66.28° at 30 min, 19.43° at 60 min, and 31.41° at 120 min. The results obtained are reflected in Tables 1 and 2 and Figures 4–7.

**Table 1.** Data obtained with the 6 W lamp.

| Group | N | Mean | Standard Deviation | 95% Confidence Interval for the Mean | |
| --- | --- | --- | --- | --- | --- |
| | | | | Lower Limit | Upper Limit |
| Control | 6 | 116.1983 | 5.89532 | 110.0116 | 122.3851 |
| 15 min | 6 | 69.6667 | 4.86648 | 64.5596 | 74.7737 |
| 30 min | 6 | 74.3833 | 3.87475 | 70.3170 | 78.4496 |
| 60 min | 6 | 73.8000 | 3.60333 | 70.0185 | 77.5815 |
| 120 min | 6 | 61.4517 | 5.85645 | 55.3057 | 67.5976 |
| Total | 30 | 79.1000 | 19.97077 | 71.6428 | 86.5572 |

**Table 2.** Data obtained with the 85 W lamp.

| Group | N | Mean | Standard Deviation | 95% Confidence Interval for the Mean | |
| --- | --- | --- | --- | --- | --- |
| | | | | Lower Limit | Upper Limit |
| Control | 6 | 117.6000 | 5.66533 | 111.6546 | 123.5454 |
| 15 min | 6 | 70.2583 | 3.70735 | 66.3677 | 74.1490 |
| 30 min | 6 | 66.2833 | 2.94715 | 63.1905 | 69.3762 |
| 60 min | 6 | 19.4333 | 1.78176 | 17.5635 | 21.3032 |
| 120 min | 6 | 31.4167 | 1.71629 | 29.6155 | 33.2178 |
| Total | 30 | 60.9983 | 35.14705 | 47.8742 | 74.1225 |

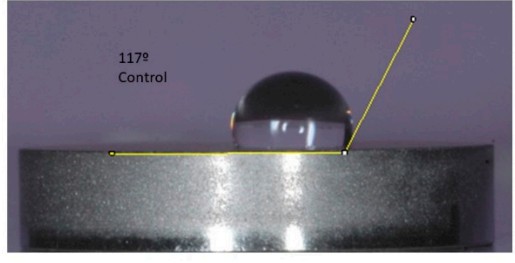
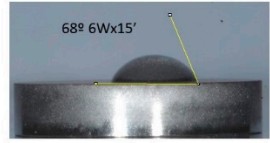
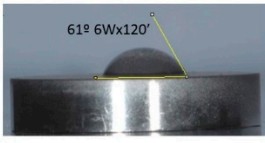
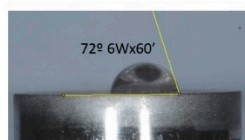
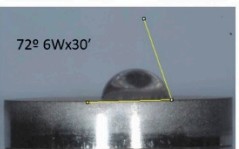

**Figure 4.** Composition of the different angles obtained with irradiation at 6 W for different time intervals compared with the control.

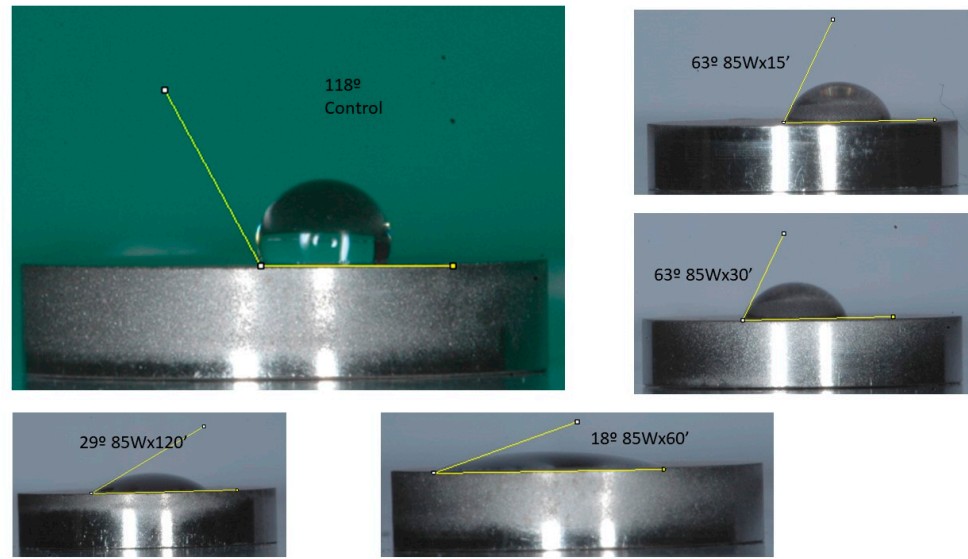

**Figure 5.** Composition of the different angles obtained with irradiation at 85 W for different time intervals compared with the control.

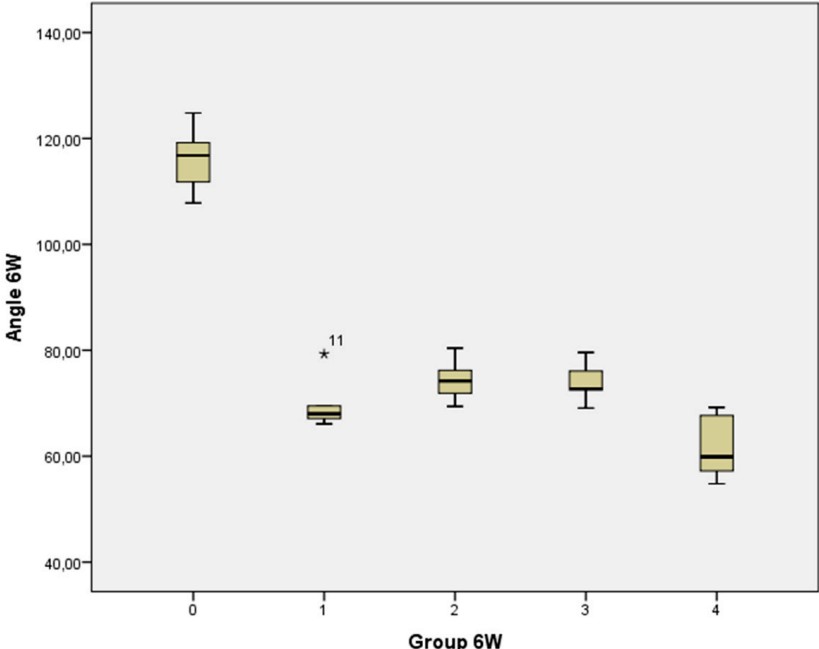

**Figure 6.** Box plot of angles obtained with a 6 W lamp at different irradiation times, including the control. The graphs show the lower (Q1), mean (Q2), and upper (Q3) quartiles.

The whiskers represent the highest and lowest values. All groups presented statistically significant differences with respect to the control. The 120 min group also showed statistically significant differences from the 30 and 60 min groups.

The whiskers represent the highest and lowest values. All the groups presented statistically significant differences with respect to the control. The 60 and 120 min groups also showed statistically significant differences from the 15 and 30 min groups.

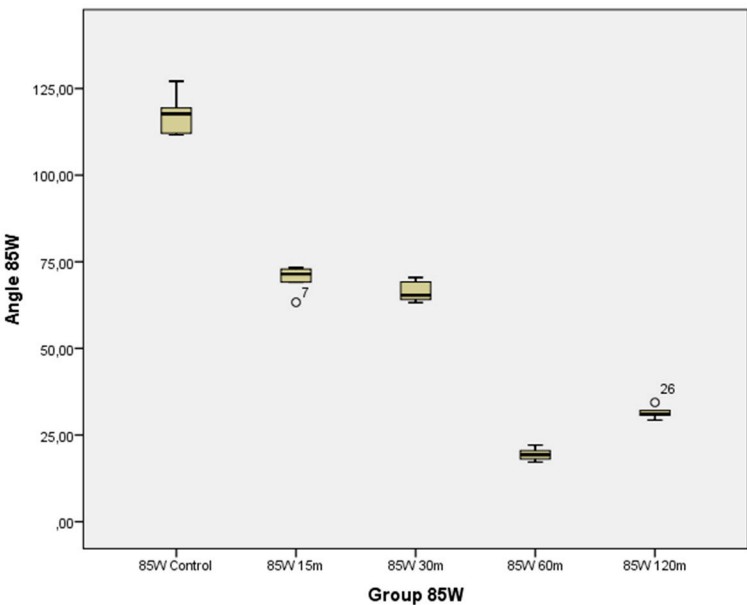

**Figure 7.** Box plot of angles obtained with an 85 W lamp at different irradiation times, including the control. The graphs show the lower (Q1), mean (Q2), and upper (Q3) quartiles.

## 4. Discussion

A perfectly clean titanium surface is considered to comprise only titanium and oxygen, which creates a titanium oxide surface. However, such pure surfaces are not available even in the laboratory, as it only a short period of atmospheric exposure is required for the surface to be contaminated with a monolayer of hydrocarbons and inorganic impurities. In a study conducted by Zhao et al. [5], lower carbon pollution was observed on surfaces that had been isolated from the surrounding environment during production compared with those that had not.

It is known that the higher the deposit of hydrocarbon compounds, the lower the hydrophilicity and the greater the angle of contact of the water. Usually, to measure this effect on hydrophilicity, the contact angle of a drop of liquid on the surface being studied is measured, and digital images of the same are analyzed.

The importance of good wettability and a low contact angle is to ensure rapid protein absorption and good cell adhesion to hinder bacterial colonization. Osteoblasts grown on hydrophilic surfaces produce more markers of differentiation by creating an osteogenic microenvironment, potentially contributing to faster and improved osseointegration.

In general, the information provided by implant manufacturers only refers to the maximum sterilization length, which is usually 5 years. However, the need for clinics to have a wide assortment of material available means that they must have an ample supply of stored implants. Due to these circumstances, it is unlikely that dental implants can be used within 4 weeks of manufacturing, implying that most implants will have already undergone a process of ageing prior to insertion.

Different methods have been proposed to preserve the surface of titanium under the best biocompatibility conditions, including packaging in ultravacuum conditions (ultrahigh vacuum), immersion in isotonic solutions, irradiation with ultraviolet C light, the use of cold plasma, and medical ozone. Whereas the first two methods attempt to keep the implant surface intact, the others focus on restoring the surface of titanium oxide, freeing it from impurities.

Perhaps the most viable and economical solution against the inevitable biological ageing of titanium is exposure to UV radiation just before use in a procedure known as photofunctionalization [6].

The underlying mechanism of photofunctionalization appears to be based on the creation of amphiphilic molecules, also called amphipathic molecules. These are molecules

that have one hydrophilic (water soluble) end and another hydrophobic end (rejects water). It was stated that modifying the electrostatic properties of the titanium surface plays a decisive role in its bioactivity. The change in surface energy is responsible for this modification. It was also mentioned that this fundamental change is caused by the formation of spaces between the oxygen atoms that allow the adhesion of OH hydroxyl radicals. Other authors attributed this effect to the energy of UVC beam photons, which induces the formation of hydroxyl radicals and hydrogen dissociation. Finally, there are authors who attributed this effect to the rupture of the C=O bonds, removing surface contaminants and regenerating a clean surface that would leave the penta-coordinated titanium (Ti5c) exposed.

Regardless of the exact mechanism, photofunctionalization creates homogeneous conditions for all implants regardless of the date of manufacture, offering the clinician the opportunity to accurately compare individual results. In this way, differences in the osseointegration of products from the same manufacturer can also be eliminated.

This effect has been observed at various UV wavelengths, including ultraviolet A (UVA), ultraviolet B (UVB), and even in combinations of both ultraviolet C light (UVC), which is commonly used due to its higher power. Two previous studies comparing titanium surfaces treated with UVA and UVC showed that despite emerging superhydrophilia on both titanium surfaces, the cellular fixation capacity was significantly greater for the UVC-treated surfaces. Reported findings indicate that photofunctionalization with ultraviolet C light accelerates the process of osseointegration, both in animals and in humans.

This effect may be of help in immediate loading procedures, as well as in short implants and in cases of poor bone quality, low primary stability, or systemic diseases [7–10].

The intention of our work was to determine the effect of both irradiation power and time on the wettability angle of standard surfaces, which does not depend solely on the composition of the material. Surface topography also plays an important role, depending on whether it is a pure titanium surface, acid-etched, or sand-blasted. Rough surfaces lose their hydrophilicity more quickly than smooth surfaces. In addition, an interesting observation is that the hydrophilicity and hydrophobicity of rough surfaces are greater than those of smooth surfaces, while their angles vary widely as a result of biological ageing.

In our study, all surfaces were of the same nature: Grade IV titanium with RBM-TC surface (Ticare, Mozo Grau, Valladolid, Spain), so that the only variables capable of affecting the wettability angle were power and time of exposure.

This study showed that UVC photoactivation modifies titanium surfaces from a hydrophobic (control) to a hydrophilic state at both 6 and 85 W, each showing statistically significant differences from the control group. However, according to the wettability classification based on the contact angle, superhydrophilic surfaces were not achieved, as no angle was less than 10°.

These differences were accentuated by an increase in time. However, the results revealed that the most significant effect depended on the power of the emitter, although the effect was not evident before 30 min. After 60 min, the effect of 85 W was significantly more pronounced (73° versus 19°) and was equally or more effective at 120 min (61° versus 31°). However, in the case of the 85 W group, this improvement did not change to any great extent, changing from a contact angle of 19° at 60 min to 31° at 120°, a paradoxical effect not seen at 6 W. We share the opinion that this is due to the saturation of compounds in the atmosphere, which, once detached from the titanium surface, are re-deposited on the titanium surface with its increased absorption capacity.

This study points to an increase in the hydrophilicity of titanium surfaces treated with UV light, and it would be of particular interest to continue with an in vivo study. Such a study should focus on applying a power of 85 W and a time of 60 min, a combination that yielded the best result in vitro.

*Study Limitations*

Some authors considered that the effect of the hydrophilicity of a given material and its impact on biological activity may be difficult to interpret [11]. However, the wide diversity of materials and variety of treatments available for implant surfaces complicate generalizing the results.

Our study is based on an in vitro model. We feel that animal studies are needed to check whether the results obtained with a power of 85 watts for 60 min produce an improvement in BIC values.

## 5. Conclusions

The exposure to UVC ultraviolet light resulted in an increase in the hydrophilicity of titanium surfaces in both groups with respect to the controls, although superhydrophilic surfaces were not obtained. Power, rather than time, was shown to be the most important and significant factor for obtaining hydrophilic surfaces.

Within the limitations of this study, we concluded that the ideal power to improve the wettability of titanium with an RBM-TC surface is 85 watts applied for 60 min at a wavelength of 254 nm. There is no improvement with longer times at this power, as the situation stabilizes and the system becomes saturated.

We consider that these findings can improve the biological process of osseointegration in implant therapy and offer us clues to the use of UVC in dental offices.

**Author Contributions:** Conceptualization, A.S.-P. and C.N.C.; methodology, N.C.-M.; investigation, N.C.-M.; data curation, M.J.M.V.; writing—original draft preparation, J.M.M.C.; writing—review and editing, A.S.-P.; N.C.-M.; C.N.C.; M.J.M.V., and J.M.M.C.; supervision, A.S.-P. All authors have read and agreed to the published version of the manuscript.

**Funding:** This research received no external funding.

**Institutional Review Board Statement:** This study was edited by American Journal Expert.

**Informed Consent Statement:** Not applicable.

**Data Availability Statement:** The data obtained in this study are available to those who request them.

**Acknowledgments:** The authors would like to thank Ticare for their collaboration in manufacturing the titanium discs.

**Conflicts of Interest:** The authors declare no conflict of interest.

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
