# Peer review of "Effect of Photofunctionalization with 6 W or 85 W UVC on the Degree of Wettability of RBM Titanium in Relation to the Irradiation Time"

_applsci, doi:10.3390/app11125427_

Round 1

Reviewer 1 Report

The topic of the article is interesting and well written.

You have to change the words "grupo" and "angle " in the box plot which are not in English 

The statistical results can be improved because they are not well define in particular when you declare the test which are used:

Tukey analysis, a Shapiro-Wilk normality test, 1-factor ANOVA and the 105 nonparametric Mann-Whitney U-test. 

On the basis of these statistical analysis it is not clear in the text whether the sample values ​​are distributed as a normal variable or not

Author Response

Comments from Reviewer 1

Thank you for your kind words.

Comment 1: You have to change the words "grupo" and "angle " in the box plot which are not in English 

Response: We agree with this comment. Therefore, have changed the words “grupo” and “angulo” to “group” and “angle” in the box plot

Comment 2: Tukey analysis, a Shapiro-Wilk normality test, 1-factor ANOVA and the 105 nonparametric Mann-Whitney U-test. 

Response: We have, change and rewritten our statistical methods 

“Statistical analysis was performed using the IBM SPSS ver. 23.0 (IBM Co., Armonk, NY, USA). A normality assumption for the data was tested using the Shapiro-Wilk test. Descriptive statistics were used to synthesize the collected data. Mean comparisons within the group were computed using analysis of variance, while the median differences in the groups were analysed by the Mann-Whitney U-test. Statistical significance was considered to be when P≤0.05”

Comment 3: On the basis of these statistical analysis it is not clear in the text whether the sample values ​​are distributed as a normal variable or not

Response: We have included a concrete sentence that stated:

“all variables followed a normal distribution except the variable 6W and 15 minutes”

Reviewer 2 Report

Esteemed colleagues,

I am happy to review your interesting manuscript and I hope that my remarks will contribute to increasing its' quality.

page 2

I would like you to elaborate on the impact of titanium ageing and its' susceptibility to contamination.

page 8

What supplementary scientific pieces of information do you consider that animal studies might bring bring to your research?

Do you plan to extend your research towards animal studies?

page 8 / 5. Conclusions

Please justify briefly why power was more important than time. factor.

Please add a final prospective clinical conclusion/application based on your results.

Kind regards.

Author Response

We would like to thank the reviewer for this positive evaluation.

Comment 1: I would like you to elaborate on the impact of titanium ageing and its' susceptibility to contamination.

Response: This is an exciting field, but given the limitations of length, we have added the following text on page 2. I am attaching an extensive comment to it.

"Over time the percentage of carbon deposited on titanium increases from 14% to 63% with important clinical consequences."

Comentary:

For a long time, it was considered that the titanium oxide surface would retain its properties. However, several papers have informed about a loss of titanium osteoconductivity over time (related with an increased period after processing the surface). The data implies time-dependent biological degradation of titanium; the phenomenon was defined as the biological aging of titanium. 1–3

It is worth noting that aged implants consistently showed lower values not only for the healing delay but also a compromised level of bone-implant integration. In both early healing stages of week 2 and later stage of week 4.4

In general, it has been considered that the percentage of contact between the bone and the implant can range from 35% to 65%.5,6 This percentage has been was primarily associated with the reduced capability of aged titanium surfaces to attract proteins and osteogenic cells. 4 X-ray photoelectron microscopy has demonstrated an increases of carbon element deposits over time. 1,7

As Att claim in 2009, “the percentage of carbon, which was 14% in the acid-etched titanium when prepared, increases with time and becomes 63% after 4 weeks of storage under an ambient condition”. 1 This deposition of hydrocarbon compounds has an inverse effect on the adhesion of proteins and the attraction of osteogenic cells.  These changes can be observed by the angle of wettability. It has been reported that titanium surface immediately manufactured has a contact angle less than 5 degrees, this angle is attenuated in 2 weeks with an angle over 40 degrees and after 4-week is more than 60 degrees. 1,2,7–9

Comment 2: page 8 What supplementary scientific pieces of information do you consider that animal studies might bring to your research?

Response and Commentary: For some authors, the degree of hydrophilicity does not correlate with the biological capabilities of titanium, such as protein adsorption and cell attraction capabilities.1,9

It is a well-known fact that the hydrophilicity of the surface is not proportional to the bioactivity of the material, with some conflicting results. 10,11

Therefore, the data from the recent studies indicates that the degree of hydrophilicity is an unlikely cause of the bioactivity level of titanium surfaces. 1,9,12,13

Comment 3: page 8 Do you plan to extend your research towards animal studies?

Response and Commentary: The welfare of the animals used in research is very important. 3R (Replacement, Reduction and Refinement) must be observed. For this reason, we developed the correct power and time of irradiation, before comparing its effects over osseointegration that include the attachment, settlement, proliferation, and differentiation of those cells, as well as the production and mineralization of extracellular matrices on titanium surfaces. Unfortunately, this environment cannot be reproduced in vitro.

Our intention is to continue with this line of research and check if power and time (85w and 60 minutes) has a positive effect on osseointegration.

Comment 4: page 8 / 5. Conclusions Please justify briefly why power was more important than time. factor.

Response: We accepted the referee’s suggestion and included the following sentence “We consider that this study can improve the biological process of osseointegration in implant therapy, and give us clues over the use of UVC in dental offices.”

Comentary:

Due to UVC being an electromagnetic radiation, its behaviour is ruled by the inverse-square law. The intensity of a light is inversely proportional to the square of the distance from the source. Due to the distance (in our study) being constant, the effect is only affected by the power of the source and time.

In our study, the effect was determined by the power more than by the time. During the same time intervals, the effect of the 85 W was more intense than the 6 W one. In consequence We reached the best combination with 85W and 60 minutes.

References:

  1. Att W, Hori N, Takeuchi M, et al. Time-dependent degradation of titanium osteoconductivity: an implication of biological aging of implant materials. Biomaterials. 2009;30(29):5352-5363. doi:10.1016/j.biomaterials.2009.06.040
  2. Hori N, Att W, Ueno T, et al. Age-dependent degradation of the protein adsorption capacity of titanium. J Dent Res. 2009;88(7):663-667. doi:10.1177/0022034509339567
  3. Iwasa F, Hori N, Ueno T, Minamikawa H, Yamada M, Ogawa T. Enhancement of osteoblast adhesion to UV-photofunctionalized titanium via an electrostatic mechanism. Biomaterials. 2010;31(10):2717-2727. doi:10.1016/j.biomaterials.2009.12.024
  4. Lee JH, Ogawa T. The biological aging of titanium implants. Implant Dent. 2012;21(5):415-421. doi:10.1097/ID.0b013e31826a51f4
  5. Weinlaender M, Kenney EB, Lekovic V, Beumer J, Moy PK, Lewis S. Histomorphometry of bone apposition around three types of endosseous dental implants. Int J Oral Maxillofac Implants. 1992;7(4):491-496. http://www.ncbi.nlm.nih.gov/pubmed/1299645.
  6. Ogawa T, Nishimura I. Different bone integration profiles of turned and acid-etched implants associated with modulated expression of extracellular matrix genes. Int J Oral Maxillofac Implants. 2003;18(2):200-210. http://www.ncbi.nlm.nih.gov/pubmed/12705297.
  7. Att W, Hori N, Iwasa F, Yamada M, Ueno T, Ogawa T. The effect of UV-photofunctionalization on the time-related bioactivity of titanium and chromium-cobalt alloys. Biomaterials. 2009;30(26):4268-4276. doi:10.1016/j.biomaterials.2009.04.048
  8. Suzuki T, Hori N, Att W, et al. Ultraviolet treatment overcomes time-related degrading bioactivity of titanium. Tissue Eng Part A. 2009;15(12):3679-3688. doi:10.1089/ten.TEA.2008.0568
  9. Hori N, Ueno T, Suzuki T, et al. Ultraviolet light treatment for the restoration of age-related degradation of titanium bioactivity. Int J Oral Maxillofac Implants. 2010;25(1):49-62. http://www.ncbi.nlm.nih.gov/pubmed/20209187. Accessed March 13, 2018.
  10. He B, Wan Y, Bei J, Wang S. Synthesis and cell affinity of functionalized poly(L-lactide-co-β- malic acid) with high molecular weight. Biomaterials. 2004;25(22):5239-5247. doi:10.1016/j.biomaterials.2003.12.030
  11. Wang YW, Wu Q, Chen GQ. Reduced mouse fibroblast cell growth by increased hydrophilicity of microbial polyhydroxyalkanoates via hyaluronan coating. Biomaterials. 2003;24(25):4621-4629. doi:10.1016/S0142-9612(03)00356-9
  12. Aita H, Hori N, Takeuchi M, et al. The effect of ultraviolet functionalization of titanium on integration with bone. Biomaterials. 2009;30(6):1015-1025. doi:10.1016/j.biomaterials.2008.11.004
  13. Ogawa T, Iwasa F, Tsukimura N, et al. TiO2 micro-nano-hybrid surface to alleviate biological aging of UV-photofunctionalized titanium. Int J Nanomedicine. 2011;6:1327. doi:10.2147/IJN.S22099